# To Gain Insights into the Pathophysiological Mechanisms of the Thrombo-Inflammatory Process in the Atherosclerotic Plaque

**DOI:** 10.3390/ijms25010047

**Published:** 2023-12-19

**Authors:** Francesco Nappi

**Affiliations:** Department of Cardiac Surgery, Centre Cardiologique du Nord, 93200 Saint-Denis, France; francesconappi2@gmail.com or f.nappi@ccn.fr; Tel.: +33-149334104; Fax: +33-149334119

**Keywords:** thromboinflammation, neutrophil extracellular traps, NETosis, ultra-large VWF, neutrophil inflammasome, COVID-19

## Abstract

Thromboinflammation, the interplay between thrombosis and inflammation, is a significant pathway that drives cardiovascular and autoimmune diseases, as well as COVID-19. SARS-CoV-2 causes inflammation and blood clotting issues. Innate immune cells have emerged as key modulators of this process. Neutrophils, the most predominant white blood cells in humans, are strategically positioned to promote thromboinflammation. By releasing decondensed chromatin structures called neutrophil extracellular traps (NETs), neutrophils can initiate an organised cell death pathway. These structures are adorned with histones, cytoplasmic and granular proteins, and have cytotoxic, immunogenic, and prothrombotic effects that can hasten disease progression. Protein arginine deiminase 4 (PAD4) catalyses the citrullination of histones and is involved in the release of extracellular DNA (NETosis). The neutrophil inflammasome is also required for this process. Understanding the link between the immunological function of neutrophils and the procoagulant and proinflammatory activities of monocytes and platelets is important in understanding thromboinflammation. This text discusses how vascular blockages occur in thromboinflammation due to the interaction between neutrophil extracellular traps and ultra-large VWF (von Willebrand Factor). The activity of PAD4 is important for understanding the processes that drive thromboinflammation by linking the immunological function of neutrophils with the procoagulant and proinflammatory activities of monocytes and platelets. This article reviews how vaso-occlusive events in thrombo-inflammation occur through the interaction of neutrophil extracellular traps with von Willebrand factor. It highlights the relevance of PAD4 in neutrophil inflammasome assembly and neutrophil extracellular traps in thrombo-inflammatory diseases such as atherosclerosis and cardiovascular disease. Interaction between platelets, VWF, NETs and inflammasomes is critical for the progression of thromboinflammation in several diseases and was recently shown to be active in COVID-19.

## 1. Introduction

Our understanding of the impact of the inflammatory process in atherogenesis and other chronic diseases has increased significantly since the pioneering work dating back over two decades [1]. Examining specific mutations that contribute to clonal hematopoiesis is crucial for understanding the process of atheromatous plaque infiltration by neutrophils. The most frequently mutated genes include DNA (cytosine-5)-methyltransferase 3A (DNMT3a), Ten-eleven-translocation 2 *(TET2),* Putative Polycomb group protein (ASXL1), and *JAK2.* A specific somatic signaling gain-of-function mutation in *JAK2* (Janus kinase 2, V617F) activates *STAT* (signal transducer and activator) and deserves emphasis. *Jak2V617F* induces marked erythrophagocytosis and neutrophil infiltration in myeloid cells [2]. The events that occur lead to an acceleration of the atherogenesis process, which is associated with an increased propensity to rupture. Additionally, two independent studies have shown that the *JakV617F* mutation is linked to both spontaneous strengthening in NET release [2] and thrombus generation [2].

Thrombotic events, including myocardial infarction after plaque rupture, are now known to involve inflammation and innate immune cells. Several studies suggest that platelet activation promotes a procoagulant state that drives inflammation and thrombosis in a vicious cycle [2,3]. Over the last decade, the idea that innate immune cells also contribute to thrombosis has gained recognition. Initially considered a safeguard response against the invasion of pathogens by means of local fibrin accumulation, it is now clear that innate immune cells can also make a significant contribution to sterile pathological thrombosis [4,5]. Thromboinflammation is now increasingly recognised as important therapeutic objective for a number of human disorders. The activation of platelets and immune cells, along with endothelial activation/dysfunction (as shown in the graphical abstract), is key to the concept of thromboinflammation. The result is microvascular thrombosis and ultimately organ dysfunction [6].

### Search Strategy

In September 2023, a review was conducted using PubMed to investigate the database with the following search terms: ‘Tromboinflammation (56 to present)’, ‘Inflammasone (2092 to present)’, ‘NET (1807 to present)’, ‘NETtosi (160 to present)’, and ‘Neutrophils coupled to NET (369 to present)’. ‘Neutrophils coupled to Inflammasome (135 to present)’, ‘Tromboinflammation coupled with Neutrophil (17 to present)’, ‘Tromboinflammation coupled with Platelet (30 to present)’, and ‘Platelet coupled with Von Willebrand Factor (15 to present)’. The search prioritised identifying data from basic research articles, reviews, observational cohort studies, and randomized controlled trials (RCTs) on the aforementioned topics. Full-text articles and reviews published in the past three years up to the end of September 2023 were included, but some commonly referenced and highly regarded older publications were not excluded. One particularity of this pathoanatomic condition is the scarcity of randomized studies available on optimal treatment. This is due to the prioritization of biomolecular research as regard this pathophysiological process.

## 2. Understanding the Role of Platelets and P-Selectin as Key Actors in Thromboinflammation

In thromboinflammation, platelets play an important role (Figure 1) [7,8,9]. Upon activation, they create heterotypic activation complexes with monocytes and neutrophils by binding to the adhesion molecule P-selectin (CD62P). P-selectin plays a critical role in leukocyte recruitment and activation and is stored and released by platelets and Weibel-Palade bodies in endothelial cells [10,11,12,13]. The interplay between P-selectin and PSGL-1 results in P-selectin being cleaved to form soluble P-selectin (sP-selectin). The substance still possesses many of its procoagulant and stimulatory characteristics [14,15,16]. High plasma levels of sP-selectin are linked to an elevated risk of cardiovascular disease, myocardial infarction, and stroke in both humans and mice [17,18,19,20].

It is noteworthy that COVID-19 patients referred for emergency cardiac surgery have experienced significant issues with acute thromboinflammation and thromboembolism, resulting in devastating post-operative complications that are difficult to manage. Similarly, patients with acute aortic dissection and NSTEMI requiring emergency revascularization have also experienced severe complications due to thromboembolic episodes before disseminated intravascular coagulation. To prevent these complications, it is crucial to closely monitor and manage thromboinflammation and thromboembolic events in these patients [21,22,23,24,25,26,27,28].

On monocytes, P-selectin binding rapidly exposes tissue factor (TF), the coagulation initiator, to the surface [29,30,31,32,33,34]. Over time, the expression of the TF gene increases and TF is released from the surface of monocytes in extracellular vesicles. The primary source of TF in the blood is then the activated monocyte [12,35]. TF and Factor VIIa combine to aid in the formation of the prothrombinase complex on the exterior of triggered platelets, thereby producing high quantities of thrombin. Interestingly, researchers [3,9] have shown that platelet-specific P-selectin deficiency alters the initiation of atherosclerosis in a preclinical model of atherosclerosis based on the work of Russell Ross [1]. This modification has the potential to decrease smooth muscle cell mobility, which may significantly impact lipid levels and the number of cells in the growing plaque [36,37,38,39]. Nevertheless, trials on patients presenting with acute coronary syndrome and administered inclacumab, an anti-P-selectin antibody, found that while the drug reduced troponin release, it did not have any effect on adverse effects during the SELECT-ACS trial [40,41]. This may be attributed to E-selectin, another type of selectin, being expressed as a result of endothelial activation through factors such as platelet factor-4 or other cytokines [42,43,44]. Perhaps in treating atherothrombosis, it is necessary to adopt a dual approach that inhibits both P- and E-selectin. Of these, 5-HT (serotonin) is of particular interest. In addition to inducing vasoconstriction, 5-HT also amplifies platelet and endothelial activation, as evidenced by its ability to facilitate Weibel-Palade body exocytosis [15,45,46]. (Etulain and colleagues proposed that platelets induce neutrophils to release neutrophil extracellular traps (NETs) via P-selectin and PSGL-1 signals during aseptic inflammation) [47].

Platelets play a multifaceted role in inflammation. These processes, along with platelet aggregation and fibrin deposition, can cause vascular occlusion even without physical injury to the blood vessels. Platelets also act as protectors by preventing bleeding in inflamed venules caused by white blood cells passing through disrupted endothelial junctions [48,49,50,51] (Figure 1).

Hook and colleagues [52] discovered that activated platelets interact with leukocytes through P-selectin glycoprotein ligand 1 (PSGL-1). Two hours after inducing systemic inflammatory response syndrome (SIRS), the expression of PSGL-1 in alveolar neutrophils was higher in gp91phox-/-mice. The interaction between platelets and neutrophils decreased in the peripheral blood of gp91phox-/-mice, indicating that activated platelets migrated to the lungs of mice lacking Nox2. Nox2-derived reactive oxygen species (ROS) play a crucial role in maintaining immune homeostasis in the lungs and resolving inflammation after a systemic inflammatory insult. The lungs are crucial organs where immune cells come into very close proximity with each other and the surroundings. Potential inflammatory stimuli may be introduced on a routine basis. The function of Nox2 in inhibiting basal platelet chemokine secretion, suppressing upregulation of neutrophil adhesion molecules and regulating a crucial neutrophil enzyme involved in the creation of NETs. Notably, tissue injury is not induced by the lack of Nox2 alone, but its absence increases the likelihood of severe injuries in organs while also creating an environment with low-level inflammation in the setting of systemic inflammation. To enhance outcomes for patients who suffer from significant inflammatory organ injury, it is crucial to take into account the diverse function of ROS in the inflammatory response [52] (Figure 2).

## 3. Interaction of NETs with Ultra-Large VWF in Thromboinflammatory Vasculopathy

### 3.1. Identifying the Role Played by NETs

An organized cell death pathway, known as NETosis, takes place in a precise subset of neutrophils in response to different pathological stimuli, such as ischemia [53]. Recently, the cell biology of NETosis has been further researched [54,55,56,57,58]. The enzyme PAD4 (protein arginine deiminase 4) plays a pivotal role in this process [56,57,58]. PAD4 possesses the sole nuclear localization signal among the PAD family. On entry into the nucleus, PAD4 converts positively charged arginine residues, common in histones, into citrulline, an uncharged amino acid. This process weakens histone interactions within nucleosomes and DNA, resulting in chromatin decondensation, histone proteolysis, and unwinding into NETs. H3 and H4 biomarkers are useful for identifying NETs in both animals and humans. They can be detected by analysing plasma samples and tissue sections [56,58].

PAD4 is thought to have specific cytoplasmic targets that impact the cell biology of NETosis and neutrophil inflammasome composition. NETosis is significantly impaired in neutrophils lacking functional PAD4, either due to genetic deficiency or inhibition [58,59,60,61]. The investigators used high-resolution time-lapse microscopy to prompt NETosis in stimulated mouse neutrophils and human neutrophil-like cells. They showed that PAD4 has the required enzymatic and nuclear localization capabilities for various stages, such as rupturing the nuclear envelope and releasing extracellular DNA [61,62]. Accordingly, researchers have provided evidence demonstrating a correlation between the elevated NETosis and heightened neutrophil PAD4 protein expression in type 1 diabetes patients [58,61]. This finding clarifies their pro-NETotic phenotype. Moreover, a critical aspect for neutrophils to undergo cell death and subsequently release chromatin involves the formation of gasdermin D-dependent membrane pores [63,64]. However, it appears that not all pathways leading to NETs result in neutrophil demise. Significantly, in the course of an infection, it was feasible to recognise well-operating cytoplasts (enucleated cells) with the ability to underpin phagocytosis [65,66].

Adorned with histones, cytoplasmic, and granular proteins, NETs create a substantial framework that damages the nearby tissue and introduces neoantigens, ultimately causing autoimmune diseases. Recently, we demonstrated that the introduction of neutrophil extracellular traps (NETs) with granulocyte colony stimulating factor (G-CSF) and collagen injections triggered arthritis and joint erosion in a mouse strain typically resistant to the disease [67]. In blood vessels, NETs, such as VWF, act as a platform for platelet adhesion and initiation of coagulation [54,55,57,58].

Active PAD4, which is released in conjunction with NETs, also facilitates the citrullination of ADAMTS13, thus impeding VWF scission and allowing platelet aggregates to remain close to the vessel wall in the coexistence of PAD4 [68,69].

Certain neutrophils release pieces of the inflammasome that include ASC (apoptosis-associated speck-like protein containing a CARD) tangled in NETs [70]. These inflammasome leftovers, when taken up by cells, have been observed to propagate inflammasome formation, similar to prion proteins. This cascade of events can increase IL-1β production and accelerate systemic inflammatory responses [71]. Studies have found TF-containing microparticles within NETs, consistent with their procoagulant function. Additionally, microparticles can originate from other cells such as malignant ones. Microparticles are derived from multiple sources including monocytes, which are the primary source of procoagulant microparticles [72]. NETosis and the increase in NET-associated TF have recently been linked to systemic inflammation and IL-1β levels, suggesting a common regulatory pathway [71]. Furthermore, activation of both canonical and non-canonical inflammasomes stimulates TF secretion from activated macrophages and monocytes, as demonstrated by recent studies [73].

Over time, a plethora of clinical and experimental evidence has linked NETs to various ischemia-related and thromboinflammatory ailments [54,55,57,58,66,74,75,76,77]. Inhibiting NET production or their cleavage through DNases has been suggested as a novel therapy, akin to ADAMTS13’s VWF cleavage [68].

Recently, Novotny and colleagues [78] conducted an in-depth histological examination of arterial thrombi, revealing differences in both thrombus architecture and leukocyte subset abundance among patients with acute ischemic stroke (AIS) and acute myocardial infarction (AMI). The researchers discovered that while leukocyte and neutrophil levels were similar between AIS and AMI thrombi, monocyte, eosinophil, B-cell, and T-cell counts were higher in stroke patients compared to those with AMI thrombi. Moreover, there was an uneven distribution of NETs in terms of quantity and appearance. These were evident in all patients with AIS, but only in 20.8% of those with AMI. The abundance of NETs in thrombi correlated with inferior outcome scores among patients with AIS. Conversely, patients with AMI displayed reduced ejection fraction. This disparity in patient outcomes distinguishes the crucial influence of NETs on thrombus stability in both disorders.

In a controlled trial of 108 acute ischaemic stroke patients, Ducrox et al. [79] reported histological findings of thrombus retrieval. The aim of the study was to investigate the presence of NETs in thrombi retrieved during endovascular therapy in AIS patients and to evaluate their impact on tissue-type plasminogen activator (tPA)-induced thrombolysis. The authors identified clusters of NETs in all thrombi. The network density was higher in the peripheral layers of the thrombus. The study has found that the presence of thrombus NET content causes resistance to reperfusion. The study investigated both mechanistic and pharmacological approaches, utilizing intravenous tPA. This was carried out irrespective of the underlying cause. Therefore, the combination of DNAse 1 with tPA should be considered a new strategy for exploration in the context of AIS. Novotny et al. [78] and Ducrox et al. [79] discuss the importance of NETs in thrombosis and their potential clinical benefits. The most significant finding is that recombinant DNAse 1 increased thrombolysis induced by tissue plasminogen activator ex vivo. However, DNAse 1 alone did not produce the same effect.

Most notably, Blasco et al. [80] presented findings of NETs in coronary thrombi among patients with COVID-19 who had STEMI. The study reveals the fundamental process of coronary blockage in STEMI patients, emphasizing the crucial involvement of NETs in the development of COVID-19-associated coronary thrombosis. Researchers found elevated levels of NETs in the blood clots of all COVID-19 patients. Specifically, patients with STEMI and COVID-19 had notably more NETs than those without, according to earlier findings from the same group. Immunohistochemical analysis demonstrated that all clots comprised a greater proportion of fibrin and polymorphonuclear cells. The complete lesion analysis, involving thrombi, NETs, and cellular infiltrate, demonstrated the absence of atheromatous plaques. In contrast, 65% of non-infected patients displayed STEMI with visible atheromatous plaques. It is also worth mentioning that the percentage of plaque fragments in the patient historical control closely resembled that of a previous series of 142 patients who did not experience STEMI [80,81]. Furthermore, it is noteworthy that in the cohort studied by Blasco et al., patients with STEMI did not report significant changes in the coagulation parameters mentioned earlier, except for one patient who had a high concentration of D-dimer. Furthermore, this investigation furnishes a reliable explanation for the significant contribution of neutrophils and NETs to coronary thrombus formation in COVID-19 subjects, despite the constraint of a small sample size [80].

Due to a lack of reliable evidence, it is unclear whether there is a causal relationship between circulating NETs and adverse clinical outcomes after STEMI. Langseth et al. analyzed serum collected an average of 18 h after PCI and correlated peripherally measured NET-specific components with clinical outcomes in STEMI [81]. The observational cohort study followed 956 patients who received PCI for STEMI for a median duration of 4.6 years. Patients’ serum double-stranded DNA (dsDNA) was used to assess the more precise NETs markers, such as myeloperoxidase DNA and citrullinated histone. The authors did not find any significant differences in the levels of NETs markers between groups with or without a primary composite endpoint that encompassed reinfarction, stroke, heart failure rehospitalization, unscheduled revascularization post the initial infarction for more than three months, or all-cause mortality, regardless of their occurrence sequence. Despite this, there was a significant increase in dsDNA levels (*p* < 0.001) in patients who didn’t survive (*n* = 76) compared to those who did. High dsDNA levels above the median were found to be associated with an increased mortality rate (54 vs. 22, *p* < 0.001), and upper quartile levels of dsDNA were linked to a greater risk of mortality. Additionally, dsDNA showed a weak correlation with D-dimer (rs = 0.17, *p* < 0.001), while elevated dsDNA levels were linked to a higher risk of all-cause mortality. Similarly, in STEMI patients, elevated dsDNA levels were weakly associated with hypercoagulability [82].

Studies by Blasco et al. [80,81] and Langseth et al. [82] have confirmed the importance of neutrophil extracellular traps (NETs) in the pathogenesis of SARS-CoV-2 infection. These results support the notion that targeting intravascular NETs is an appropriate goal in the management of patients with STEMI and represents a viable method to prevent coronary thrombosis in patients with severe COVID-19 [77,80,81] (Figure 3).

### 3.2. The Role of VWF and ADAMTS13

Von Willebrand factor (VWF) is a multimeric glycoprotein that binds to platelet glycoprotein Ibα and plays a crucial role in the recruitment and activation of platelets. Von Willebrand factor (VWF) is located in the same area as P-selectin, found in Weibel-Palade bodies of ECs and α-granules. It plays a key role in supporting platelet tethering and leukocyte adhesion in a similar fashion. Moreover, if the ultra-large VWF stored in Weibel-Palade bodies is not cleaved, it forms long strings, which are temporarily anchored to the endothelial surface [83]. In artificial endothelial microchannels, von Willebrand factor (VWF) released from activated endothelial cells associates with itself to form elongated strands that can span across the vascular lumen [84,85]. These ultra-large VWF molecules fragment red blood cells, leading to the formation of schistocytes, as observed in thrombotic thrombocytopenic purpura.

ADAMTS13 is mainly produced in the liver. Its primary role is to cleave von Willebrand factor (VWF) anchored on the endothelial surface, in circulation, and at the sites of vascular injury. A deficiency of plasma ADAMTS13 activity (<10%) caused by mutations of the ADAMTS13 gene or autoantibodies against ADAMTS13 leads to hereditary or acquired (idiopathic) TTP. ADAMTS13 activity is typically normal or only slightly reduced (by more than 20%) in other forms of thrombotic microangiopathy that are secondary to hematopoietic progenitor cell transplantation, infection, disseminated malignancy, or hemolytic uremic syndrome. Currently, plasma infusion or exchange is the preferred initial treatment. However, new treatments such as recombinant ADAMTS13 and gene therapy was developed. ADAMTS13 deficiency is a risk factor for developing myocardial infarction, stroke, cerebral malaria, and preeclampsia [84].

Thrombotic thrombocytopenic purpura impairs the activity of ADAMTS13 [84], a metallopeptidase that contains a thrombospondin motif type 1 member 13) [86,87]. Fractured red blood cells release heme, which triggers NETosis, intensifying the thromboinflammatory process [88,89]. The uncut VWF, with its activated binding sites, stimulates the creation of platelet strings (see Figure 4) and microthrombi [90,91]. In cardiovascular disease and stroke, elevated concentrations of VWF are linked to an increase in the severity of the disease [92,93,94]. Studies in mice show that deficiency of ADAMTS13, an enzyme that converts VWF, results in increased thrombosis and inflammation [95,96]. The same group has published two reports that show how recombinant ADAMTS13 can potentially reduce inflammation in mice suffering from stroke and myocardial ischaemia/reperfusion injury. The investigators have shown that recombinant ADAMTS13 has a protective anti-inflammatory effect when administered in both scenarios [97,98].

Histological examination of both animal and human thrombi reveals interwoven fibrin, NETs, and VWF localized in the solid matrix of thrombi. VWF stabilises the thrombus by acting as a bridge between the vessel wall and the NETs. NET-associated histone released from the NET enhances VWF delivery from ECs and stimulation of platelet activation [83,99]. Figure 4 demonstrates a clear interaction between VWF and NETs. DNA and histones both bind to VWF, effectively anchoring NETs in place [100,101,102]. In a crucial study [103] predating the discovery of NETs [57], a distinct binding between the A1 domain of VWF and histones had been identified, elucidating the purpose of this interaction site on VWF. It has been observed that recombinant ADAMTS13 therapy not only clears VWF from endothelial surfaces, but also removes NETs [83,84,104,105,106,107] (Figure 4).

### 3.3. Inflammasone to the Direction of All Actors

Excluding cellular interactions, the thromboinflammatory process is complex. It involves the interaction of the coagulation cascade, complement and cytokines, predominantly the IL-1 family. A variety of regulatory proteins are produced as inactive precursors and necessitate proteolytic processing in order to attain biological functionality. Initially reported in 1989 by Black et al. [108,109], caspase 1, which is predominantly accountable for processing pro-IL-1β intracellularly [110], was eventually unveiled as the primary component of the inflammasome [111,112]. Inflammasomes are complex protein structures composed of several components that gather in innate immune cells after activation [113]. Recent reports suggest that inflammasome assembly in neutrophils, which was previously mainly researched in monocytes/macrophages, also takes place (Figure 4). Although neutrophils are responsive to the same stimuli as monocytes, the latest studies have demonstrated that they do not require LPS priming in vitro, indicating a more rapid response time, aligning with their function as the host’s primary immune defense [60]. The main driving force for the assembly of the inflammasome in sterile thrombo-inflammation is the activated platelet [114,115]. Please refer to Figure for further details. It should be noted that there are numerous other intracellular and extracellular stimulators that activate different pathways of activation, which are discussed in more detail [116,117]. The importance of the inflammasome in pro-IL-1β processing renders it a focal point in the evolution of thromboinflammation, thereby presenting a plausible target for modifying inflammatory pathways. The most extensively studied inflammasome in IL-1β activation is the NLRP3 receptor (pyrin domain containing three of the NLR family) [118].

Once activated and released, IL-1β acts as a pro-inflammatory cytokine, inducing the production of endothelial adhesion molecules that attract leukocytes, including E-selectin, ICAM-1 (intercellular adhesion molecule-1) and VCAM-1 (vascular cell adhesion molecule-1) [119,120,121,122,123]. The importance of the inflammasome in controlling the excessive pro-inflammatory effects caused by the release of IL-1β is highlighted in some rare genetic disorders. Gain-of-function mutations in the NLRP3 inflammasome are responsible for cryopyrin-associated periodic syndromes (CAPS). Canakinumab, an anti–IL-1β antibody, is approved as an orphan medication for treating these conditions. A recent study showed that an exclusively neutrophil-expressed mutation that induces inflammasome assembly is sufficient to trigger cryopyrin-associated periodic syndromes. This discovery emphasizes the vital function of neutrophils in cryopyrin-associated periodic syndromes [112]. Thiam and colleagues [62] have recently discovered that the formation of ASC specks is a useful indicator of inflammasome activity in murine neutrophils, occurring before chromatin decondensation. In line with this, Münzer and colleagues [60] demonstrated that NLRP3 deficiency considerably decreases NETosis in vitro and leads to a lower density of NETs in thrombi created by a mouse model of deep vein thrombosis induced by stenosis. This is an intriguing discovery as it places the inflammasome upstream of yet another thromboinflammatory process known as NETosis. Additionally, citrullination promotes inflammasome assembly [124], and in neutrophils, the citrullinating enzyme is PAD4 [76,125].

Pathological thromboinflammation is underpinned by the progression of the PAD4-mediated inflammasome in neutrophils and subsequent NETosis. PAD4-mediated inflammasome assembly and subsequent NETosis underpin pathological thromboinflammation, connecting the immunological role of neutrophils to the activation of platelets and monocytes (Figure 5).

## 4. Focusing on the Thromboinflammation Process in Atherosclerosis and COVID-19

It is evident that the interaction between platelets, VWF, NETs, and inflammasomes plays a crucial role in the progression of thromboinflammation associated with various diseases. In this text, we will explore two distinct pathological processes where thromboinflammation is believed to be a contributing factor.

### 4.1. Implication of Atherosclerosis

In 2019, it will become possible to target inflammation in atherosclerosis using clinical intervention, as shown by CANTOS, which supports the thesis that the development and progression of atherosclerosis is mainly due to the inflammatory response [1,119,120,121,122,124,126]. The outcomes of this significant study demonstrate that interventions aimed at inflammation can lead to encouraging clinical results. The CANTOS trial evidenced that administering an anti-IL-1β antibody to patients with cardiovascular stability after a myocardial infarction, following the guidelines, lowered the occurrence of renewed major adverse cardiovascular events [113]. Nevertheless, this constructive result was associated with a considerable increase in infections, some of which resulted in fatalities [113]. In this section, we examine the interconnection between atherogenesis and thromboinflammation while identifying significant contributors. The goal is to identify potential therapy targets. See other sources for a comprehensive overview of the condition [126,127].

As with other thromboinflammatory illnesses, in atherosclerosis, the initial stages of leukocyte recruitment are reliant on the activation of endothelium and platelets [128]. Platelets secrete chemokines (such as CCL5) and cytokines (IL-1 beta) which promote monocyte/neutrophil adhesion to the endothelium [129]. Leukocyte adhesion and the progression of atherosclerotic lesions are supported by both platelet and endothelial P-selectin [38]. Soluble E-selectin increases the risk of cardiovascular disease [17,130,131]. In mice, the absence of both P-selectin and E-selectin had the greatest negative impact on lesion progression [131,132]. A possible role for VWF in initiating platelet function within the lesion has been suggested by ultrasound molecular imaging, which has shown increased VWF-mediated platelet adhesion to the endothelium preceding the development of atherosclerotic plaques [133]. This finding explains that VWF-deficient mice fed with a high-fat diet developed atherosclerotic-prone sites later than their wild-type counterparts, and in unique locations [134].

In turbulent flow where lesions tend to form, von Willebrand factor (VWF) is necessary to facilitate platelet adhesion, ultimately marking the site for monocyte recruitment. In addition, the fatty streaks observed in VWF-deficient mice were smaller in size and contained fewer monocytes. This finding is in line with the discovery that endothelial VWF, as opposed to platelet-derived VWF, is essential in the development of atherosclerosis in mice [135]. In all animal disease models where VWF is pathologically released, ADAMTS13 deficiency accelerates the progression of atherosclerotic lesions, in contrast to VWF deficiency [136,137].

High plasma levels of sP-selectin are correlated with the severity of cardiovascular disease in humans, similar to VWF. Knock-in mice were created by deleting the cytoplasmic domain necessary for P-selectin storage, resulting in an enhanced procoagulant state due to the overproduction of thrombin. These mice exhibited an increased susceptibility to atherosclerosis [20]. Although monocytes are widely considered the most critical factor in atherosclerosis induced by lipids, there is evidence to support the involvement of neutrophil-derived NETs in a process similar to endothelial erosion in mice [126]. Recent genetic research revealed the presence of such NETs in this disease, with PAD4 deficiency being an effective inhibitor. Reducing endothelial discontinuity and EC apoptosis in a mouse model was also possible with the administration of DNAse [138]. This treatment not only tackles the blood clotting effects caused by NETs but also their toxicity. Additionally, it has been discovered that histone H4 in neutrophil extracellular traps (NETs) can cause the death of smooth muscle cells in the arterial wall, which accelerates the destabilisation of atherosclerotic plaques [139]. This finding may explain how acute infections, causing an excessive amount of NETosis, contribute to cardiovascular risk [126,140,141]. Targeting NETs could be beneficial for enhancing plaque stability in secondary prevention therapy, such as for carotid disease and stable coronary artery disease [126].

The feedback loop between inflammasome assembly and NETosis via PAD4 promotes IL-1 beta activation, which promotes NETosis and contributes to atherosclerosis [142]. There is strong experimental evidence for the activation of the NLRP3 inflammasome by cholesterol crystals and thus a direct link to the path of atherogenesis. Research has indicated that reducing atherosclerosis in mice can be accomplished by inhibiting NLRP3 or genetically deleting it [126,143,144,145]. Additionally, as NLRP3 deficiency also decreases NETosis and the accumulation of NETs in thrombi [125], inhibition of PAD4 and NLRP3 targeting may present potential interventions for hindering atherothrombosis (Figure 6).

### 4.2. Implication of Thromboinflammation in COVID-19

The World Health Organization is responsible for collecting and distributing mortality statistics. Nature Journal investigators have been monitoring the COVID-19 pandemic since the start of 2020. Reported COVID-19 mortality statistics are unreliable in many countries due to differences in testing availability, varying diagnostic capabilities, and inconsistent certification of COVID-19 as the cause of death. Apart from its direct impact, the pandemic has caused significant collateral damage resulting in loss of lives and livelihoods. Investigators recently reported a comprehensive and consistent measurement of the impact of the COVID-19 pandemic by estimating excess deaths, by month, for 2020 and 2021. The investigators have used an overdispersed Poisson count framework that applies Bayesian inference techniques to quantify uncertainty to predict all-cause deaths during the pandemic period in locations where complete reported data is lacking. Investigators have estimated that there were 14.83 million excess deaths globally, which is 2.74 times higher than the 5.42 million deaths reported as being due to COVID-19 during the same period. There are significant variations in the estimates of excess deaths among the six regions of the World Health Organization [147].

Since the COVID-19 outbreak and the frequent occurrence of coagulopathy, thromboinflammation has been attracting attention. Blood clots in dialysis and ECMO circuits have been observed in COVID-19 patients despite adequate anticoagulation, in line with the definition of thromboinflammation [145]. The principal pathogenic mechanisms underlying the pathology of COVID-19 are endothelial dysfunction [2,55,57,58,146,148,149] and thromboinflammation [125]. SARS-CoV-2 infection involves the vascular endothelium, leading to endotheliitis, thrombosis, and infiltration of inflammatory cells. In severe cases, the vascular barrier ruptures and edema occurs [2,26,55]. The resulting vascular angiogenesis, either intussusceptive or germinative, distinguishes the pulmonary pathobiology of COVID-19 patients from those with severe influenza virus-related infections [24]. Immunohistochemistry showed that ACE2 was highly expressed in alveolar epithelial and endothelial capillary cells in the autopsy findings of patients who died from severe forms of COVID-19. Additionally, ACE2-positive lymphocytes were present, and evidence of an interaction between the ACE2 receptor and immune cells was detected in the perivascular tissue or the alveoli of the lungs of patients infected with SARS-CoV-2 [55,57,58]. SARS-CoV-2 infection can cause acute inflammation, which may destabilise atherosclerotic plaques and lead to acute myocardial infarction (AMI). The cytokine storm, characterised by the release of cytokines such as IL-1α, IL-1β, IL-6, and TNF-α, is thought to play a significant role in SARS-CoV-2 infection. These cytokines can disrupt the protective functions of the normal endothelium and exacerbate pathological processes. Self-induction of the pro-inflammatory cytokine IL-1 is closely linked to the pathophysiological mechanism of cytokine storm. IL-1 induces its own gene expression, resulting in an amplification of its production levels and leading to a cytokine storm [2,55,57,58].

Additionally, IL-1 induces the expression of other proinflammatory cytokines, such as TNF-α and IL-1, and leukocyte invasion. The latter stimulates the production of chemotactic molecules, such as chemokines, which cause the infiltration of inflammatory cells into the tissues [2,55,57,58].

Patients with the disease present an increase in platelet activation, formation of platelet-monocyte aggregates, and a significant rise in monocyte tissue factor expression [150]. Despite the use of antiplatelet therapy, the outcome of hospitalized COVID-19 patients has not shown any significant improvement [151,152,153]. This suggests that severe COVID-19 is not solely driven by platelet aggregation but rather by the interaction between platelets and various factors, such as NETs. Patients with COVID-19 exhibit a specific coagulopathy marked by heightened levels of fibrinogen, D-dimer fibrin degradation products, and significantly increased levels of VWF in the bloodstream [153,154,155]. There is emerging evidence that that heightened activation of endothelial cells, causing the release of large von Willebrand factor (VWF) multimers, in combination with insufficient VWF cleavage due to ADAMTS13 consumption or disease pathophysiology related to COVID-19, might lead to escalated interactions between platelets and vessel walls, ultimately resulting in thrombotic microangiopathy [156]. Reports reveal COVID-19 to cause an excessive stimulation of the complement and coagulation systems [156,157,158].

Due to its procoagulant nature, it’s not unexpected that sP-selectin is related to disease severity in COVID-19 patients [159,160]. However, a placebo-controlled, randomised trial evaluating the impact of monotherapy dose of crizanlizumab, a P-selectin inhibitor, in mild COVID-19 patients found no substantial improvement in clinical outcomes compared to placebo, despite evidence of reduced thrombin activation and sP-selectin levels [161]. It is planned to conduct a larger study with a larger number of random doses. Furthermore, neutrophil extracellular traps (NETs) are probably implicated in the thromboinflammation responsible for COVID-19 and the subsequent severe lung impairment [162]. In 2012, scientists revealed NETs’ pathological involvement in acute respiratory distress syndrome associated with acute lung injury due to blood transfusion in mice and humans, with NETs found in the lungs [163,164]. Similarly, autopsy case reports of COVID-19 patients have identified NETs in the lung parenchyma and alveolar space [165]. Aymonnier et al. [166] conducted a study of freshly isolated neutrophils from severe COVID-19 patients as part of a clinical trial on DNase 1 inhalation (https://www.clinicaltrials.gov, accessed on 15 December 2023; unique identifier: NCT04402944). Investigators discovered that a significant number of neutrophils were poised for NETosis, evidenced by 40% of nuclei testing positive for citrullinated histone H3. Additionally, 2% of neutrophils taken from either blood or tracheal aspirate (i.e., from the lung) were observed to be forming inflammasome, as detected by ASC speck assembly. Observation of the spots in the vicinity of multilobulated neutrophil nuclei before the standard nuclear rounding of NETosis indicates that inflammasome assembly takes place prior to NETosis in COVID-19 patients [166], thereby highlighting its potential as a treatment target. In our study, though, neutrophils and monocytes depicted a comparable rate of inflammasome positivity (2%). However, it is imperative to note that the bloodstream contains ten times more neutrophils than monocytes. Thus, the possible underestimation of the importance of neutrophil-produced IL-1β cannot be ignored. The inflammasome activates an autocrine IL-1β feed-forward loop that may be involved in the pathogenesis of adult acute respiratory distress syndrome, cytokine storm and microvascular thrombosis, ultimately leading to multi-organ failure in severe COVID-19 cases [166] (Figure 7).

## 5. Nex Steps

It is evident that thrombosis and inflammation do not occur in isolation during diseases. As further knowledge is acquired, the complexity of vascular processes is unveiled. For example, the formation of a blood clot can no longer be interpreted as being solely a matter of cleaved fibrinogen with plasma proteins, since the DNA content of the clot also has to be taken into account for clot lysis. Likewise, platelet clots are no longer just fibrinogen-crosslinked platelets. It was unexpected to discover that thrombi could still be produced in mice following an injury, despite their lack of both fibrinogen and VWF [167].

Recent studies show that deep vein thrombosis formation is not only related to platelets and red blood cells, but also involves monocytes and neutrophils [168,169,170]. Researchers have been aware of the role of platelets and platelet-derived factors in inflammation since the beginning with ongoing research supporting their impact [2,3,7,32,102]. The traditional cellular players of inflammation and thrombosis fully intertwine, validating the term thromboinflammation.

Neutrophils have a central role in thromboinflammation since they can assemble inflammasomes and produce NETs. NETs contribute equally to thrombosis, immune response, and tissue damage, making them critical to the development of this disease. NETosis sets off platelets, leads to the release of toxic histones [101,171,172,173] and active enzymes, hence modifying thrombus stability [174,175]. The assembly of the inflammasome, along with heightened PAD4 activity, leads to the discharge of cytokines that are both pro-inflammatory and pro-thrombotic. This also activates the endothelium, further increasing the recruitment of leukocytes as well as inflammasome assembly in adjacent cells. As such, PAD4 and inflammasome emerge as pertinent targets for antithromboinflammatory therapy. However, there are numerous outstanding mechanistic queries that remain unanswered. Identifying the specific protein targets of PAD4 citrullination necessary for inflammasome formation, and the intracellular substrates of caspase 1 (the enzyme produced by inflammasomes) that must be cleaved to trigger NETosis, could enhance our comprehension of how to regulate the behaviour of neutrophils.

It is important to note that the function of PAD4 is multifaceted. The role of PAD4-induced NETosis via cathepsin G-mediated platelet-neutrophil interaction in ChAdOx1 vaccine-induced thrombosis has recently been investigated. Patients with VITT show increased thrombogenesis through PAD4-mediated NET formation via cathepsin G-mediated platelet-neutrophil interaction [176].

Some aspects require consideration. The implementation of therapeutic techniques, ongoing clinical trials, and the potential contribution of Netosis inhibitors may offer avenues for further investigation into NET and NETosis. The expression of cytokine Chi3l1 has recently been investigated in relation to the role of certain mediators that significantly inhibit Netosis [176]. In the context of triple-negative breast cancer (TNBC), the restriction of CD8+ T cells in the stroma is associated with adverse clinical outcomes and a lack of response to immune-checkpoint blockade (ICB). To identify factors causing T cell stromal restriction, murine breast tumours that lacked the transcription factor Stat3 has been analyzed. Stat3 is commonly hyperactive in breast cancers and promotes an immunosuppressive environment. The expression of the cytokine Chi3l1 was reduced in Stat3-/-tumours. CHI3L1 expression was higher in human TNBCs, and other solid tumours that showed T cell stromal restriction. Ablation of Chi3l1 in the polyoma virus middle T model of breast cancer generated an immune response against tumours and postponed the onset of mammary tumours. These outcomes corresponded with an elevated T cell infiltration to the tumour and an enhanced response to immune checkpoint blockade. In terms of mechanisms, Chi3l1 facilitated neutrophil recruitment and extracellular trap formation, resulting in the inhibition of T cell infiltration. Taifour et al., sheds light on the mechanism that limits CD8+ T cell function in stromal tissue and proposes that targeting Chi3l1 may enhance anti-tumor immunity across numerous types of tumors [177].

Finally, an ongoing randomized study, currently in phase 1, is evaluating the effect of danirixin on NETs in individuals suffering from chronic obstructive pulmonary disease (COPD). This mechanistic study intends to evaluate the ability of danirixin to decrease the formation of NETs, also known as NETosis. The subjects were randomly allocated in a 3:1 ratio to either 35 mg of danirixin hydrobromide taken orally twice a day or a placebo for 14 days. Participants can use inhaled COPD maintenance and rescue medication throughout the study. The study involves a screening period with a maximum duration of 30 days, followed by a 2-week treatment period, and a 1-week phone call follow-up appointment. Approximately 50 participants will undergo screening, of whom around 24 will successfully complete the study [178,179] Table 1. 

## 6. Limitation

One limitation of the study is its reliance on observational data, which may introduce biases in study design. Additionally, the exclusion of non-English studies, particularly those from China, the initial epicenter of the COVID-19, may have limited the scope of the study. In addition, the limited availability of randomized studies on optimal treatment for patients with thromboinflammatory disease limits the validity of this review.

## 7. Conclusions

Clearly, thrombosis and inflammation are not isolated diseases. As we gain more knowledge, the complexity of vascular processes becomes more apparent. For instance, the formation of a blood clot cannot be solely attributed to cleaved fibrinogen and plasma proteins, as the DNA content of the clot must also be considered for clot lysis. Similarly, platelet clots do not consist solely of platelets cross-linked with fibrinogen. Despite the absence of both fibrinogen and VWF, thrombi can still form in mice following an injury.

Recent studies have shown that the formation of deep vein thrombosis is not only associated with platelets and erythrocytes. Monocytes and neutrophils are also involved. Several investigations have highlighted the role of platelets and platelet-derived factors in inflammation. Ongoing research continues to support their impact. Inflammation and thrombosis are closely linked, supporting the term thromboinflammation.

Due to their newly discovered ability to form inflammasomes and produce NETs, neutrophils have been identified as central to thromboinflammation. NETs play a critical role in the development of this disease by contributing equally to thrombosis, immune response and tissue damage. NETosis triggers platelets, resulting in the release of toxic histones and active enzymes, which modifies thrombus stability. Inflammasome assembly, together with increased PAD4 activity, results in the release of cytokines that are both proinflammatory and prothrombotic. This activates the endothelium, which in turn increases the recruitment of leukocytes and the assembly of inflammasomes in neighboring cells. Therefore, potential targets for antithromboinflammatory therapy include PAD4 and the inflammasome. Nevertheless, many mechanistic questions remain to be answered. Our understanding of how to regulate neutrophil behaviour may be improved by identifying the specific protein targets of PAD4 citrullination required for inflammasome formation and the intracellular substrates of caspase 1 (the enzyme produced by inflammasomes) that need to be cleaved to induce NETosis.

## Figures and Tables

**Figure 1 ijms-25-00047-f001:**
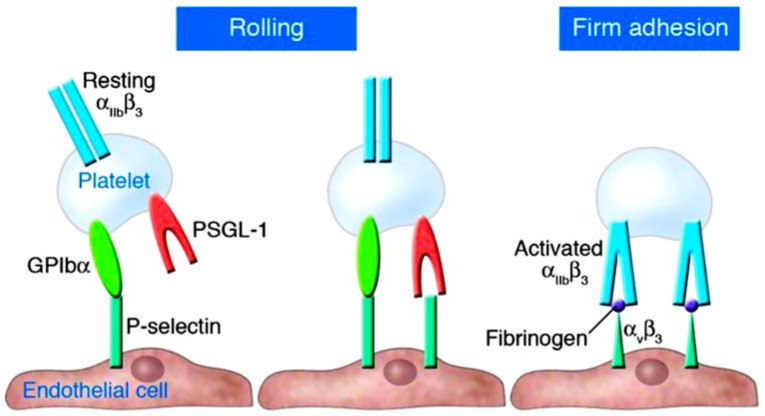
Platelet-endothelium adhesion occurs when activated endothelial surfaces express P-selectin, which interacts with platelet surface receptors GPIbα and PSGL-1 to mediate platelet rolling. Beta-3 integrins subsequently mediate firm adhesion. From Gawaz et al. Ref. [8].

**Figure 2 ijms-25-00047-f002:**
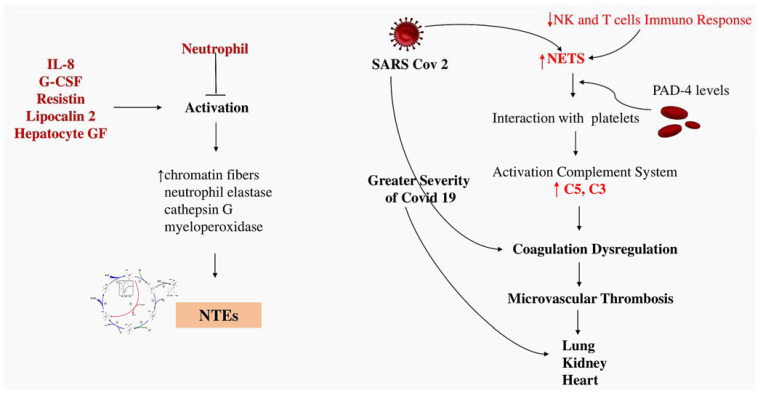
Neutrophil activation is mediated by several factors including IL-8, G-CSF, resistin, lipocalin-2, hepatocyte growth factor and NET re-release. The immune responses of both NK and T lymphocytes contribute to the formation of NETs, which in turn activate the complete system (C5 and C3). This leads to microvascular thrombosis and subsequent organ damage. Abbreviations: C, complement; GF, growth factor; IL, interleukin; NK, natural killer. Other abbreviations are listed in the previous figure. From Ref. [26].

**Figure 3 ijms-25-00047-f003:**
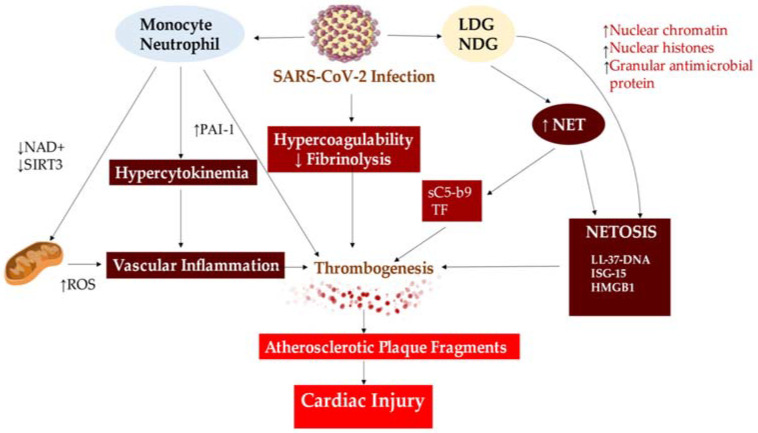
The formation of Neutrophil Extracellular Traps (NETs) in severe COVID-19 patients leads to cardiac injury caused by vascular inflammation, thrombogenesis, and NETOSIS, which arise from the unstable atherosclerotic plaque. Abbreviations used: HMGB1, High Mobility Group Box 1; ISG-15, Interferon-Stimulated Gene 15; LDG, Low-Density Granulocytes; NDG, Normal Density Granulocytes; NAD, Nicotinamide Adenine Dinucleotide; ROS, Reactive Oxygen Species; SIRT3, Sirtuin 3. The same abbreviations are used as in the previous figure. The symbol ↑ represents an increase, while ↓ represents a decrease. From Ref. [57].

**Figure 4 ijms-25-00047-f004:**
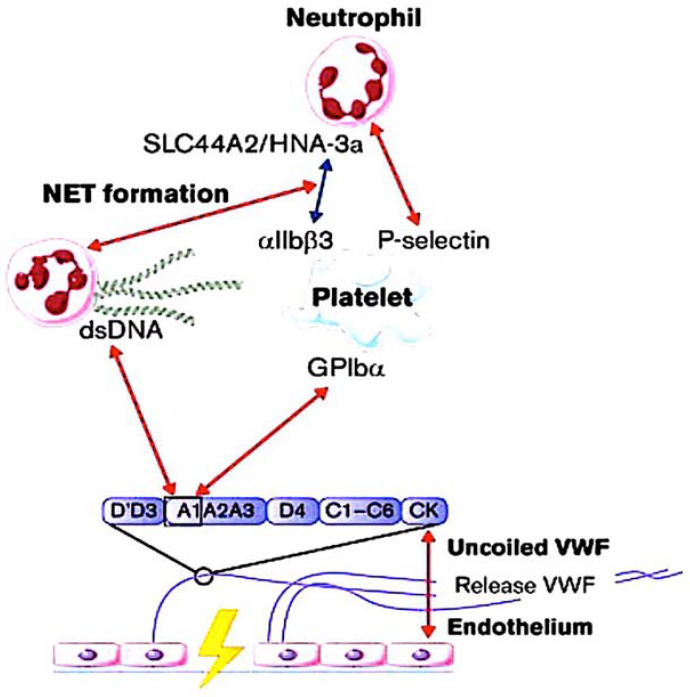
Flow models were used to investigate interactions between von Willebrand factor (VWF) and neutrophils, providing insights into the relationships between the A1 domain of VWF multimers, platelets, neutrophils, and NETs under conditions of high shear flow (indicated by red arrows) and low shear flow (indicated by blue arrows). Abbreviations; ds, double strand; GP, glycoprotein.

**Figure 5 ijms-25-00047-f005:**
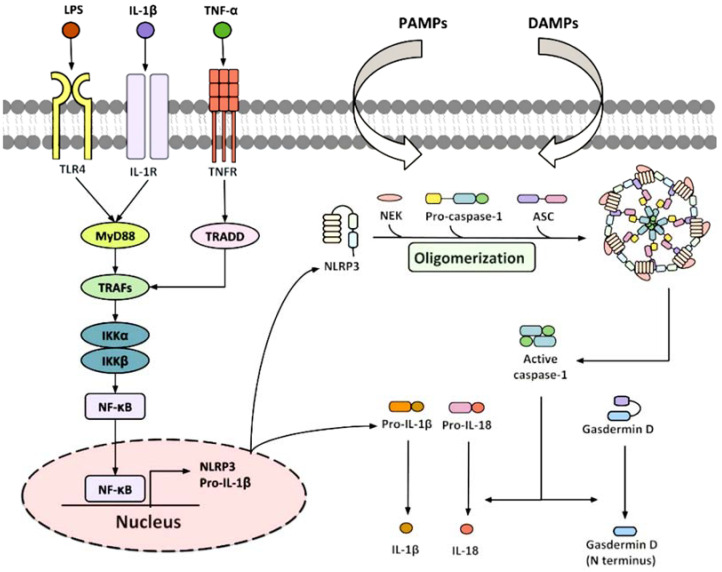
Typical signalling pathways for NLRP3 inflammasome activation. Upon stimulation of TLR4, IL-1R, or TNFR, TNF receptor-associated factor 2 (TRAF2) and TNF receptor-associated factor 6 (TRAF6) recruit the inhibitor of nuclear factor-κB kinase α/β (IKKα/β), resulting in the translocation of NF-κB subunits to the nucleus. This enhances the transcription of NLRP3 and pro-IL-1β, facilitating the activation of components in the local systems. Upon NLRP3 inflammasome initiation through PAMPs and DAMPs, the subunits of NLRP3 and pro-IL-1β are enabled. As a result, the inactive procaspase-1 is cleaved into active caspase-1, which triggers the following assembly of NLRP3 inflammasome. Once activated, the NLRP3 inflammasome initiates the processing of signals. Gasdermin D, pro-IL-1β, and pro-IL-18 are transformed into their biologically active forms through the cleavage of dormant procaspase-1 into active caspase-1, which then triggers the processing of gasdermin D, pro-IL-1β, and pro-IL-18 into their biologically active forms. From Wu et al. Ref. [123].

**Figure 6 ijms-25-00047-f006:**
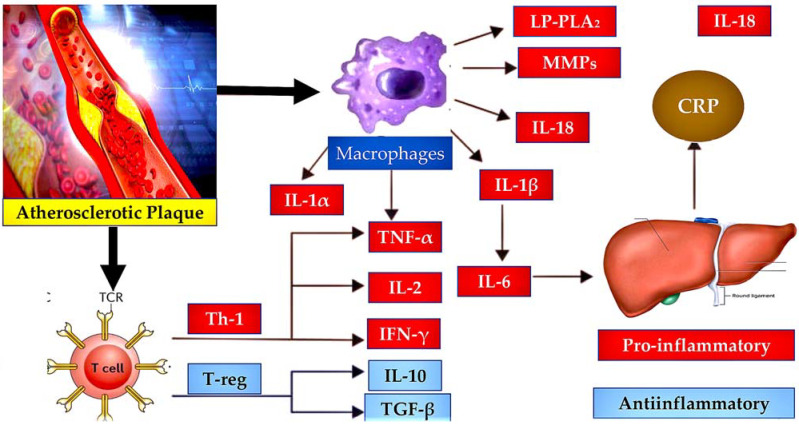
Inflammatory pathways implicated in atherosclerosis are illustrated. Preclinical and clinical trials have revealed a complex equilibrium between pro-inflammatory and anti-inflammatory pathways. The balance is regulated by inflammatory cells (macrophages and T-cells) and the liver (CRP), promoting endothelial dysfunction and the progression of atherosclerotic plaque via the production of molecules with either a pro-inflammatory (red box) or anti-inflammatory (blue box) effect. Plaque rupture is a potential result of an intensified inflammatory process. Abbreviations used: CRP (C-reactive protein); MMPs (matrix metalloproteinases); IFN-γ (interferon-gamma); IL-1α (interleukin-1-alpha); IL-1β (interleukin-1-beta); IL-2 (interleukin-2); IL-6 (interleukin-6); IL-10 (interleukin-10). IL-18, interleukin-18; Lp-PLA2, lipoprotein-associated phospholipase A2; TGF-β, transforming growth factor beta; Th-1, T-helper-1 lymphocyte; TNF-α, tumor necrosis factor alpha; and T-reg, regulatory T lymphocyte. From Ref. [146].

**Figure 7 ijms-25-00047-f007:**
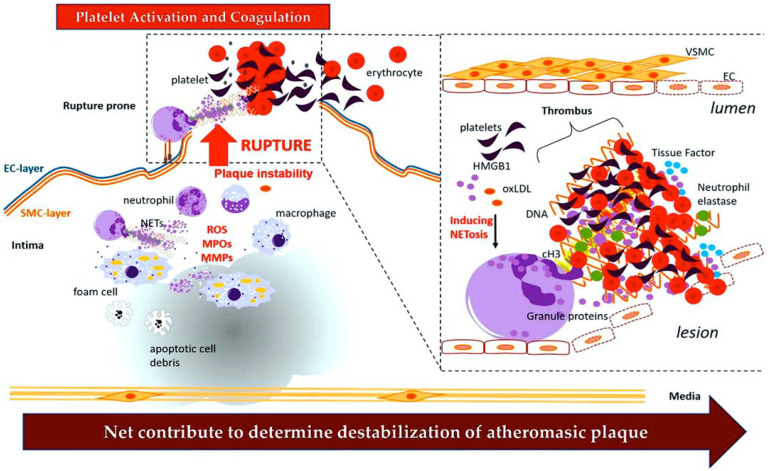
NETs cause arterial blockage. During atherothrombosis, NETs produced by neutrophils can contribute to a range of biochemical events that can activate the coagulation process. By destabilising the atheromatous plaque and inducing its rupture, NETs enhance the stability of the blood clot. From Ref. [2].

**Table 1 ijms-25-00047-t001:** Clinical Studies on Mediators of Thromboinflammation with Therapeutic Applications.

Therapeutic Agent/Drug Name	Indication	Study No./Status	Preclinical/Clinical Study Outcome	Refs.
Crizanlizumab	P-selectin inhibitor	NCT03814746	No improvement in clinical outcomes	[161]
Pulmozyme	Reduces NETosis in neutrophils.	NCT04402944	Phase 2	[166]
Targeting Chi3l1	Advanced cancer. Enhance anti-tumor immunity	“ ” “ ”	Reduced levels of helper and cytotoxic T cells in TNBC tumors led to decreased tumor infiltration.	[177]
Danirixin hydrobromide	Reduces NETs formation	NCT02453022	Phase 1	[178,179]

## Data Availability

Not applicable.

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
