# Peer review of "To Gain Insights into the Pathophysiological Mechanisms of the Thrombo-Inflammatory Process in the Atherosclerotic Plaque"

_ijms, 2023, doi:10.3390/ijms25010047_

Round 1

Reviewer 1 Report

Comments and Suggestions for Authors

This review described the existing knowledge on the cellular processes involved in immuno-thrombosis, focusing on the role of neutrophil extracellular traps (NETs), platelets coagulation, and inflammatory pathways. This topic is very interesting since understanding underlying mechanisms in immuno-thrombosis has become important for developing more efficient therapies to treat and prevent thrombosis in disorders associated with thrombotic risk immuno-mediated. However, the topic is not unexplored and in addition, it describes how the deregulation of hemostasis is associated with pathological conditions using only COVID-19 as a clinical model.  So, I have some suggestions. In particular:

1.     In addition to COVID-19, Why have no other pathologies involving immuno-thrombosis been described? for example Antiphospholipid syndrome (APS) or VITT? if you do not intend to discuss other pathologies, I suggest including COVID-19 information in the abstract for greater clarity.

2.     P-selectin can drive both Netosis and ROS formation via NADPH oxidase. Introduce and discuss this issue. Moreover, in Figure 1, I suggest amplifying and deepening the endothelium activation as it is strongly associated with platelet activation.

3.     Lines 145-150: I suggest removing p value in the text.

4.     Platelet-neutrophil interaction via cathepsin G is a mechanism that mediates the release of NETs (Carnevale et al. ATVB 2023). This important aspect has been completely missed in this review.

5.     In my opinion, it is necessary to add tables to describe the main characteristics and results of the pre-clinical and clinical data discussed in the text.

6.     Are there currently used therapeutic strategies? Are there clinical trials? are they efficient and safe? Netosis inhibitors, for example. I believe that this issue should be discussed at least in the conclusions.

7.     A suggest adding a representative figure that describe paragraph 4.

8.     In my opinion the title does not reflect the focus of the review

Minor comment:

1.     Keywords are missing

2.     add abbreviations to the first mention in the text: line 160 (tPA), line 296 (ASC)

3.     the references in the text do not match the list. (80,81,85,86 etc.)

Author Response

The author thanks the reviewer for the reported comments. Responses are indicated below marked in red The author thanks the reviewer for the reported comments. Responses are indicated below marked in red 

1. In addition to COVID-19, Why have no other pathologies involving immuno-thrombosis been described? for example Antiphospholipid syndrome (APS) or VITT? if you do not intend to discuss other pathologies, I suggest including COVID-19 information in the abstract for greater clarity.

I 'agree with the reviewer on the importance of Antiphospholipid syndrome (APS) and VITT. However, this review was designed with the purpose of discussing the pathogenetic mechanisms of the thrombo-inflammatory process in the atherosclerotic plaque. In particular, this investigation was designed after the increase of STEMI cases in patients who had had a Covid 19. 
The reviewer's suggestion to mention the two topics in the abstract was fulfilled

2.     P-selectin can drive both Netosis and ROS formation via NADPH oxidase. Introduce and discuss this issue. Moreover, in Figure 1, I suggest amplifying and deepening the endothelium activation as it is strongly associated with platelet activation.

Corrections have been included in the text a figure (Figure 2) has been added. 

3.     Lines 145-150: I suggest removing p value in the text.

P value have been removed 

4.     Platelet-neutrophil interaction via cathepsin G is a mechanism that mediates the release of NETs (Carnevale et al. ATVB 2023). This important aspect has been completely missed in this review.

Thank you for the pertinent observation. However, the study by Carnevale et al is focused on PAD4-Induced NETosis Via Cathepsin G-Mediated Platelet-Neutrophil Interaction in ChAdOx1 Vaccine-Induced Thrombosis-Brief Report. VITT was not evaluated in this review.
A consideration was reported in the conclusion. 

5.     In my opinion, it is necessary to add tables to describe the main characteristics and results of the pre-clinical and clinical data discussed in the text.

Thank you for this important comment. The paragraph has been included

6.     Are there currently used therapeutic strategies? Are there clinical trials? are they efficient and safe? Netosis inhibitors, for example. I believe that this issue should be discussed at least in the conclusions.

Thank you for this key suggestion. A paragraph has been added in the conclusion.

7.     A suggest adding a representative figure that describes paragraph 4.

Two figures were added (Figure 6 & 7)

8.     In my opinion the title does not reflect the focus of the review

The title has been changed

Minor comment:

1.     Keywords are missing

Keywords have been included

2. add abbreviations to the first mention in the text: line 160 (tPA), line 296 (ASC)

The abbreviations have been checked

3. the references in the text do not match the list. (80,81,85,86 etc.)

The references have been checked

Reviewer 2 Report

Comments and Suggestions for Authors

1. There are some grammatical, alignments and typographical errors are noted in the manuscript and it should be thoroughly checked and corrected throughout the manuscript. For example,

·         in line number 16, the words “important” may be as “an important”;

·         in line number 55, the words “Covid 19” may be as “COVID-19”;

·         in line number 292, “showing” as “showed”;

·         in line number 361, “overproduction” as “the overproduction”;;

·         in line number 363, “lipid” as “lipids”;

·         in line number 426, “blood stream” as “bloodstream”.

The grammar mistakes and typos which are not mentioned here are also to be checked and corrected properly.

2. Check the abbreviations throughout the manuscript and introduce the abbreviation when the full word appears the first time in the abstract (The use of abbreviations in the abstract section may distract readers who wish to quickly skim through several publications before deciding to read one in full. It may therefore help to write out terms fully in this section, for example, COVID-19, CVD) and the remaining for the text and then use only the abbreviation (For example, tissue factor (TF), von Willebrand factor (VWF), etc.,). Make a word abbreviated in the article that is repeated at least three times in the text, not all words to be abbreviated.

3. The literature search should be described in detail. The authors are encouraged to include the database, search engines (like PubMed, ScienceDirect, Google scholar etc.,), the keywords used etc., which may be included since it is a review article.

4. The initial cited with reference in the text should be removed and should be in the author instruction of the journal (For example, “Gawaz M et al”) and it should also be checked all over the manuscript.

5. In the title “4.2. Implication of Thromboinflammation in COVID-19”, the authors are encouraged to include more data about COVID-19 since this part appears less informative about COVID-19, thus this section should be indicated as detailed to understand the manuscript in clear.

6. The authors may cite recent prevalence or incidence data or death rate about COVID-19 under the heading “4.2. Implication of Thromboinflammation in COVID-19” and it should be at-least of October or November 2023.

7. The reference cited in the conclusion section should be removed and it may be given in any other part of the manuscript. The conclusion should be key points of the overall observation of the review only and not with others.

8. The limitation of the present review may be given along with conclusion or under separate heading for understanding the concepts  clearly.

Author Response

The author thanks the reviewer for the comments

1. There are some grammatical, alignments and typographical errors are noted in the manuscript and it should be thoroughly checked and corrected throughout the manuscript. For example,

- in line number 16, the words "important" may be as "an important";

- in line number 55, the words "Covid 19" may be as "COVID-19."

- in line number 292, "showing" as "showed";

- in line number 361, "overproduction" as "the overproduction";;

- in line number 363, "lipid" as "lipids";

- in line number 426, "blood stream" as "bloodstream."

The grammar mistakes and typos which are not mentioned here are also to be checked and corrected properly.

The corrections are shown in yellow in the text. Regarding line 16 the entire abstract has been rewritten

2. Check the abbreviations throughout the manuscript and introduce the abbreviation when the full word appears the first time in the abstract (The use of abbreviations in the abstract section may distract readers who wish to quickly skim through several publications before deciding to read one in full. It may therefore help to write out terms fully in this section, for example, COVID-19, CVD) and the remaining for the text and then use only the abbreviation (For example, tissue factor (TF), von Willebrand factor (VWF), etc.,). Make a word abbreviated in the article that is repeated at least three times in the text, not all words to be abbreviated.

Abbreviations have been removed from the 'abstracte and have been checked in the text

3. The literature search should be described in detail. The authors are encouraged to include the database, search engines (like PubMed, ScienceDirect, Google scholar etc.,), the keywords used etc., which may be included since it is a review article.

A paragraph has been included

 4. The initial cited with reference in the text should be removed and should be in the author instruction of the journal (For example, "Gawaz M et al") and it should also be checked all over the manuscript.

The correction has been made and the text has been checked

5. In the title "4.2. Implication of Thromboinflammation in COVID-19," the authors are encouraged to include more data about COVID-19 since this part appears less informative about COVID-19, thus this section should be indicated as detailed to understand the manuscript in clear.

The author thanks the reviewer for the comment. The text has been implemented as suggested by the reviewer.

6. The authors may cite recent prevalence or incidence data or death rate about COVID-19 under the heading "4.2. Implication of Thromboinflammation in COVID-19" and it should be at-least by October or November 2023.

The author thanks the reviewer for the comment. The text has been implemented as suggested by the reviewer.

7. The reference cited in the conclusion section should be removed and it may be given in any other part of the manuscript. The conclusion should be key points of the overall observation of the review only and not with others.

The conclusion and new steps chapter has been revised. The conclusion has been rewritten as suggested by the reviewer

8. The limitation of the present review may be given along with conclusion or under separate heading for understanding the concepts clearly.

The limitation has been described in a separate paragraph

Reviewer 3 Report

Comments and Suggestions for Authors

1) Keywords are missing.

2) How atherosclerotic plaques are developed? Give molecular basis of investigation behind that.

3) Note that acute thrombo-embolism was a significant issue in Covid 19 patients referred for emergency cardiac surgery, with devastating post-operative complications that were difficult to manage. Modify the sentence.

4) However, it is important to note that platelets have a complex role in inflammation. How?

5) They can be detected by analysing plasma samples and tissue sections. Is it so?

6) Immunohistochemical analysis demonstrated that all clots comprised a greater proportion of fibrin and polymorphonuclear cells. Correct it.

7) There are numerous Grammatical errors. Correct those.

8) Improve the Figure 2

9) How to do ADAMTS13 study?

10) Neutrophils have been identified as central to thromboinflammation, owing to their 446

newly-discovered ability to assemble inflammasomes and produce neutrophil extracellu- 447

lar traps (NETs). Check the sentence.

Comments on the Quality of English Language

Correct English language throughout the text

Author Response

The author thanks the reviewer for the comments

1) Keywords are missing.

Keywords have been included 

2) How atherosclerotic plaques are developed? Give molecular basis of investigation behind that.

The change was included as suggested by the reviewer. (Line 38-48 marked in yellow)

3) Note that acute thrombo-embolism was a significant issue in Covid 19 patients referred for emergency cardiac surgery, with devastating postoperative complications that were difficult to manage. Modify the sentence.

The modification was made. (Line 71-77 marked in yellow)

4) However, it is important to note that platelets have a complex role in inflammation. How?

The change was made. (Line 100-103 marked in yellow)

5) They can be detected by analyzing plasma samples and tissue sections. Is it so?

This whole part of the paragraph has been changed (Line 100-130)

6) Immunohistochemical analysis demonstrated that all clots comprised a greater proportion of fibrin and polymorphonuclear cells. Correct it.

The text has been revised

7) There are numerous grammatical errors. Correct those.

The text has been improved

8) Improve the Figure 2

Figure 2 has been revised 

9) How to do ADAMTS13 study?

The study has been discussed (line 286-295). 

10) Neutrophils have been identified as central to thromboinflammation, owing to their newly-discovered ability to assemble inflammasomes and produce neutrophil extracellular traps (NETs). Check the sentence.

The sentence has been revised (Line 540-542).

Round 2

Reviewer 1 Report

Comments and Suggestions for Authors

The paper was substantially improved and most questions have been answered.

However, some issues need to be answered before publication:

1) Sorry but I think there was a misunderstanding. My previous suggestion was to mention COVID-19 in the abstract, not VITT and APS. Please, organize the abstract again.

2) The preclinical and clinical data discussed in the text are not yet summarized in a table, in my opinion, this could be useful for the reader.

3) line 231…Novotny 78)…: there is a parenthesis missing before the reference.

4) Finally, the references in the text once again do not match the list: Ex.: Line 235, Blasco et al. (80); line 222: “Ducrox et al. (79)” this author is not in the reference list as well as at line 210: “Novotny and colleagues” (78); Etc. Please check and correct the references.

Author Response

The author thanks the reviewer for the comments

1) Sorry but I think there was a misunderstanding. My previous suggestion was to mention COVID-19 in the abstract, not VITT and APS. Please organize the abstract again.

The abstract has been revised

2) The preclinical and clinical data discussed in the text are not yet summarized in a table, in my opinion, this could be useful for the reader.

Table 1 has been included

3) line 231...Novotny 78)...: there is a parenthesis missing before the reference.

The correction has been inserted

4) Finally, the references in the text once again do not match the list: Ex.: line 235, Blasco et al. (80); line 222: "Ducrox et al. (79)" this author is not in the reference list as well as at line 210: "Novotny and colleagues" (78); Etc. Please check and correct the references.

The references have been revised